# Lookahead Branching for Neural Network Verification

## Abstract

In this paper, we investigate the effect of lookahead branching strategy in neural network verification. We present a general recipe for integrating lookahead into any branch-and-bound search framework, and also describe how in addition to guiding branching, lookahead can generate additional lemmas that accelerate verification. We instantiate the method in two representative branch-and-bound-based verifiers (Marabou and $\alpha$-$\beta$-CROWN), and demonstrate consistent reductions in overall verification time across both systems.

## 1 Introduction

Deep neural networks (DNNs) have become state-of-the-art solutions in various domains (Sallab et al., 2017; Mnih et al., 2013; He et al., 2015). To ensure the deployment of DNN-based systems in safety critical domains, there has been significant effort from the formal methods and machine learning communities on developing scalable formal verification techniques that can rigorously reason about the behaviors of a neural network (Katz et al., 2017; Singh et al., 2019; Zhang et al., 2018; Wang et al., 2021). State-of-the-art complete neural network verifiers are based on branch-and-bound (BaB), which involves performing case splitting on non-linear activation functions (e.g., ReLU) and analyzing the cases using an incomplete verifier; if the analysis by the incomplete verifier is inconclusive, further case splits are recursively performed.

The branching heuristic, the strategy to choose which case to split on, has significant impact on verification efficiency. Ideally, the branching heuristic should lead to easier subproblems. Failing to do so, especially at the top of the search tree, can result in duplicated verification efforts and increased verification time. In the past decades, a number of branching heuristics have been developed for neural network verification (Katz et al., 2017; Wu et al., 2020; Bunel et al., 2020; De Palma et al., 2021). By and large, existing branching heuristics aim to make a decision *quickly* by leveraging local information (e.g., variable bounds) gathered during the solving process to estimate the effect of branching on certain neurons. While the branching decision is made rapidly, this approach increases the risk of ineffective branching. This is because the decision is made without evaluating the long-term impact of splitting on a neuron. Recent studies have shown it is beneficial to spend more effort on deciding the next branching neuron, by actually performing branching on a number of candidate neurons and evaluating its effect (De Palma et al., 2021; Botoeva et al., 2020).

In this paper, we systematically study this alternative branching strategy that spends *significant effort* on selecting the next neuron to split. We adopt the terminology in formal methods and call this approach *lookahead* (Heule & van Maaren, 2009). We present a general recipe for constructing a lookahead-based branching strategy for a BaB neural network verification procedure. Lookahead involves simulating potential branching decisions by performing multiple splits to measure downstream effects. By analyzing the impact of each simulated branch, lookahead uses more information to make the branching decision. We present lookahead as a template algorithm with several tunable parameters such as the preselect strategy, lookahead depth, and evaluation metric. These parameters can be instantiated in a solver-specific manner. In addition to informing the branching decision, lookahead might also discover new information, such as tightened variable bounds. We show this information can be used to derive valid lemmas at the current search level, and potentially prune the search space. We instantiate lookahead in two distinct BaB-based complete verification tools, Marabou (Katz et al., 2019; Wu et al., 2024) and $\alpha$-$\beta$-CROWN (Wang et al., 2021; Xu et al., 2020;

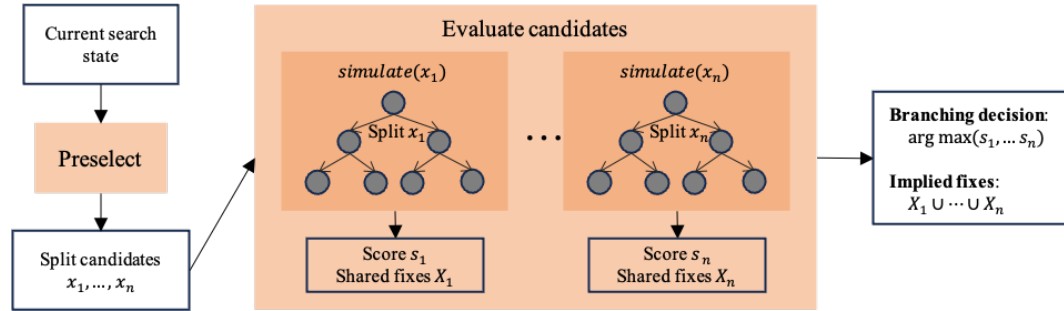

Figure 1: Visual overview of the lookahead procedure

Zhang et al., 2022a; Zhou et al., 2024), and demonstrate that lookahead can consistently boost the performance of both tools.

An overview of the lookahead workflow is presented in Figure 1. At a given search state when branching is required, a set of split candidates is pre-selected and lookahead simulates branching on each candidate up to a certain depth. The outcome of each simulation is a score along with a number of tightened bounds, which might entail that certain currently unstable neurons are fixed to particular activation phases. In the end, lookahead outputs the next neuron to split on and a set of lemmas discovered during the lookahead simulations.

The rest of the paper is organized as follows: Section 2 provides relevant background on neural network verification. Section 3 presents the general lookahead approach and concrete instantiation in Marabou and $\alpha$-$\beta$-CROWN. Section 4 provides an evaluation of lookahead branching, comparing it to existing heuristics in Marabou and $\alpha$-$\beta$-CROWN. We review related work in Section 5. Finally, we conclude and discuss future directions in Section 6.

## 2 BACKGROUND

### 2.1 NEURAL NETWORK VERIFICATION

For a trained deep neural network $N : \mathbb{R}^n \to \mathbb{R}^m$ with an input $x \in \mathbb{R}^n$ and output $y = N(x) \in \mathbb{R}^m$, the DNN verification problem is whether or not there exists an input $x$ that produces an output $y$ that satisfies a property $\phi(y)$. If there exists an input $x$ that leads to an output $y$ that satisfies the property $\phi(y)$, then the problem is satisfiable, denoted SAT. If there exists no input $x$ that can produce an output $y$ that satisfies $\phi(y)$, then the problem is unsatisfiable, denoted UNSAT.

### 2.2 BOUND PROPAGATION

To refine the search space of a neural network verification problem, bound propagation (Singh et al., 2019) estimates activation ranges at each layer, defining upper and lower bounds for each neuron. The Rectified Linear Unit (ReLU) activation function is defined as $\text{ReLU}(x) = \max(0, x)$. A ReLU neuron is considered active if its lower bound is strictly greater than zero ($\underline{z}^{(l)} > 0$), meaning its output will always be its input. Conversely, a ReLU neuron is inactive if its upper bound is less than or equal to zero ($\overline{z}^{(l)} \leq 0$), meaning its output will always be zero. Otherwise, the activation status is not yet known and the ReLU in unstable. If a ReLU's bounds can guarantee the neuron is always active or inactive, unnecessary computations can be eliminated. Bound propagation is run repeatedly in the BaB procedure to refine the search space. Through this iterative process, the bounds on neuron activations are progressively tightened, leading the solver closer to a solution.

### 2.3 BRANCH-AND-BOUND

The branch-and-bound (BaB) framework, illustrated by Figure 2, is an efficient approach to neural network verification. BaB systematically tightens the bounds of a neural network by splitting on unstable ReLU neurons. When an unstable ReLU is split, the problem is turned into two problems, one where the ReLU is active and one where the ReLU is inactive. After bound propagation, this turns a big problem into two more manageable problems. To verify with BaB, ReLU splits are repeatedly applied until each resulting subproblem can be definitively classified as either satisfying the property (SAT) or not satisfying it (UNSAT).

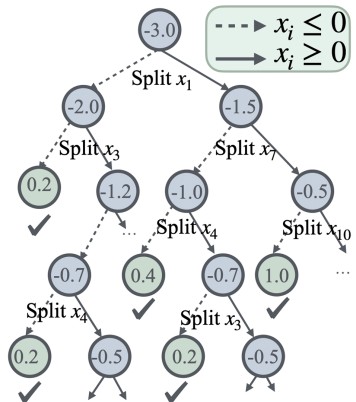

The choice of ReLU to split on is imperative for the efficiency of BaB, as it determines how fast a solver converges to a solution; well-selected splits make significantly more progress to a solution than poorly-chosen splits. Thus, heuristics to select ReLUs to split on are essential to efficient neural network verification. By and large, branching heuristics make decisions on local information like neuron bounds and pre-activation values without considering their long-term impact, often leading to inefficient search paths. We introduce lookahead branching, a novel heuristic inspired by SAT-solving techniques that strategically evaluates potential branching decisions before choosing a constraint to branch on.

Figure 2: Each node encodes a subproblem from BaB by splitting unstable ReLUs. Green nodes are pruned; blue nodes require further branching. (Zhou et al., 2024) .

## 3 METHODOLOGY

In this section, we first present lookahead as a template algorithm and discuss its properties. We then discuss concrete strategies for instantiating the lookahead procedure.

### 3.1 LOOKAHEAD FOR BAB

Algorithm 1 formalizes the lookahead branching procedure. We assume that the procedure is invoked in a branch-and-bound search shell when a branching decision is needed. At a high level, the algorithm takes as input the set of unstable neurons at the current search node and returns the neuron to branch on, along with variable bounds discovered during lookahead. The procedure depends on a number of parameters: 1) $\mathbf{d}$ is the depth of lookahead; 2) PRESELECT takes as input a collection of neurons $R$, and choose a subset of $R$ as candidates to examine closer; 3) BASESELECT is the heuristic that selects the next neuron to branch on during the lookahead simulation; 4) EVALUATESTATE evalutes the current search state and returns a real-valued score $S$, as well as sound lower- and upper-bounds of variables at the current state; and 5) AGGREGATESCORE aggregates the evaluation scores collected during lookahead simulation.

The algorithm begins by identifying a set of unstable neurons to simulate splits on. Given that performing a full lookahead simulation on every unstable ReLU is computationally expensive, a preselect strategy is employed to shrink the candidate set. The preselect strategy differs based on implementation as detailed in Section 3.2.

For each preselected candidate, the algorithm simulates the effect of branching on it with the SIMULATESPLIT sub-routine. The SIMULATESPLIT function simulates a case split on the neuron, creating one branch for each of the activation phases. For the ReLU activation function, there would be two phases (active and inactive). The algorithm then recursively explores the next level of the search tree by selecting another neuron to split on, using the BASESELECT heuristic. This process continues until a specified lookahead depth ($\mathbf{d}$) is reached. The goal of performing additional splits is to explore the potential cascading effects of each candidate split, as the effect of splitting on one neuron might not be fully realized in the immediate subproblem. After each split is simulated, the solver performs bound propagation. This is important, as it tightens the bounds of other neurons in the network based on the simulated split. When the lookahead depth is reached, the EVALUATESTATE function is called to evaluate the current state of the solver and collect the

---

**Algorithm 1** Lookahead Branching

---

1: **Input:** Set of unstable neurons $R$ at the current search level
2: **Output:** $\langle r', l, u \rangle$ where $r'$ is the neuron to split, $l$ and $u$ are sound lower- and upper-bounds of variables at the current search level
3: **Parameters:** lookahead depth **d**, preselect strategy PRESELECT, explore strategy BASESELECT, evaluation strategy EVALUATESTATE, and aggregate strategy AGGREGATESCORE
4: **function** LOOKAHEAD($R, d$)
5:     $R' \leftarrow$ PRESELECT($R$)                                    ▷ Pre-select candidate neurons
6:     $A \leftarrow \{\}$
7:     $\mathcal{L}, \mathcal{U} \leftarrow [], []$
8:     **for each** $r \in R'$ **do**
9:         $S', l, u \leftarrow$ SIMULATESPLIT($r, \mathbf{d}, R/\{r\}$)          ▷ Simulate branching and evaluate
10:        $A[r] \leftarrow$ AGGREGATESCORE($S'$)        ▷ Aggregate the scores from subproblems
11:        $\mathcal{L}$.append($l$), $\mathcal{U}$.append($u$)
12:     **return** $\arg\max(A)$, elementwise-max($\mathcal{L}$), elementwise-min($\mathcal{U}$)
13: **function** SIMULATESPLIT($r, d, R$)
14:     **if** $d == 0$ **then**
15:        $s, l, u \leftarrow$ EVALUATESTATE()
16:        **return** $[s], l, u$
17:     **else**
18:        storeSolverState()
19:        $S, \mathcal{L}, \mathcal{U} \leftarrow [], [], []$
20:        **for each** $p \in$ phases($r$) **do**
21:           applySplit($p$)
22:           propagateBounds()
23:           **if** $d == 1$ **then** $r' \leftarrow$ BASESELECT($R$)
24:           **else** $r' \leftarrow nil$
25:           $S', l, u \leftarrow$ SIMULATESPLIT($r', d-1, R/\{r'\}$)
26:           $S \leftarrow S :: S'$
27:           $\mathcal{L}$.append($l$), $\mathcal{U}$.append($u$)
28:           restoreSolverState()
29:        **return** $S$, elementwise-min($\mathcal{L}$), elementwise-max($\mathcal{U}$)

---

current variable bounds. Importantly, the solver state is saved and restored after each simulated split to ensure that the lookahead simulation does not interfere with the actual search process. This allows for independent evaluation of each candidate neuron without side effects from other simulated branches. In the end, the SIMULATESPLIT function returns the score of each simulated leaf, as well as the loosest lower and upper bounds of the variables discovered during the simulation. Importantly, these bounds are sound lower and upper bounds of the variables at the current search level.

The scores from all simulated branches are aggregated using the specified aggregation strategy to produce a single score for each candidate neuron. The neuron with the best aggregated score is selected as the next split. Moreover, the tightest lower and upper bounds discovered during lookahead are returned, which can be used to further prune the search space.

In general, lookahead can be computationally expensive, as it requires repeated simulation of the solving process. In practice, it is beneficial to invoke Algorithm 1 at the top of the search tree, and fall back to more efficient branching heuristics later on. In the next section, we discuss the different strategies for preselecting neurons, evaluating simulated branches, and aggregating scores.

### 3.2 PRESELECT STRATEGIES

If lookahead were to be performed on every neuron in a large neural network, the computational overhead would outweigh the improvement in branch quality. Thus, it is essential to preselect a subset of neurons on which to run lookahead. Several preselect strategies are possible.

One approach is to use a polarity-based strategy, that uses a heuristic that scores neurons based on the sensitivity of their activation status to changes in their bounds. For example, a score such as $\max\left(\frac{ub}{lb}, \frac{lb}{ub}\right)$, where $ub$ and $lb$ are the upper and lower bounds of a neuron, can be used to identify neurons whose activation status is most undecided. Neurons with the highest scores are then selected for lookahead. We used this polarity-based preselect strategy in Marabou.

Another approach is to use existing branching scores, such as BaBSR scores, to rank neurons. In this case, a fixed number of neurons with the highest scores are chosen for lookahead.

Regardless of the specific strategy, it is important that the preselect method identifies promising candidates using lightweight metrics, ensuring that only the most relevant neurons are considered for the more expensive lookahead procedure.

## 3.3 SCORING FUNCTIONS

The scoring function is a central component of lookahead branching, as it quantifies the effectiveness of each simulated split and guides the branching decision. Two general classes of scoring metrics are commonly used: neuron-fixing-based metrics and bound-reduction-based metrics.

**Neuron-fixing-based metric.** This approach evaluates a split by counting how many previously unstable neurons become phase-fixed (i.e., their activation status is determined) after bound propagation in each branch. To encourage both high total progress and balanced outcomes, a balance score is computed for each split. For example, if a split results in $a$ neurons fixed in one branch and $b$ in the other, the score can be defined as $\frac{a \times b}{a+b+1}$. This formula favors splits that not only fix many neurons overall but also distribute the fixes more evenly between branches. Consider a split that fixes 9 neurons in one branch and 1 in the other. Its score would be $\frac{9 \times 1}{9+1+1} \approx 0.8$. In contrast, a split that fixes 5 neurons in each branch would yield a score of $\frac{5 \times 5}{5+5+1} \approx 2.3$. Our metric would then favor the more balanced outcome in case the same number of neurons are fixed. When lookahead is performed to greater depths, the scores are computed recursively: at the leaves, the score is based on the number of phase fixes, and at each internal node, the balance formula is applied to the scores of its child branches, propagating upward to yield a final score for each candidate split. This polarity-based metric was used in Marabou.

**Bound-reduction-based metric.** This metric is based on the reduction in variable bounds or other continuous measures of progress, such as the width of neuron bounds or their proximity to a decision threshold. For example, a scoring function may combine the width of a neuron's bounds, its distance from zero, and an estimate of its influence on the objective (e.g., via gradient approximation). In a lookahead setting, after simulating a split, the scoring function can aggregate the scores from subsequent branches using a balance formula similar to the neuron-fixing metric. For instance, if $S^+$ and $S^-$ are the scores from the two branches after a split, the lookahead score can be defined as $S_0 + \lambda \cdot \frac{S^+ \times S^-}{S^+ + S^- + 1}$, where $S_0$ is the immediate score for the split and $\lambda$ is a discount factor to control the influence of deeper lookahead. For deeper lookahead, this aggregation is applied recursively as scores are propagated up the lookahead tree. This bound-reduction-based metric was used in $\alpha$-$\beta$-CROWN.

Regardless of the specific metric, an effective scoring function should capture both the potential for maximal progress toward a solution and the balance of outcomes across different branches. This ensures that the branching decision not only accelerates convergence but also avoids highly imbalanced splits that may lead to inefficient search.

## 3.4 PHASE FIXING VIA LOOKAHEAD SPLITS

One significant outcome of the lookahead branching procedure is its ability to refine variable bounds, which can lead to phase fixing of previously unstable neurons. Specifically, during the lookahead process, bound propagation is applied after simulating splits, and the resulting bounds can sometimes determine that a neuron is stable (i.e., its activation phase is fixed).

The following theorem formalizes this effect:

**Theorem 1** (Phase Fixing via Lookahead). *Let $z = \mathrm{ReLU}(y)$ be an unstable ReLU, i.e., $\ell_y < 0 < u_y$. Suppose we simulate a split on another unstable ReLU $z_r = \mathrm{ReLU}(y_r)$, generating two subproblems corresponding to the inactive ($y_r \leq 0$) and active ($y_r \geq 0$) cases.*

*Let $\ell_y^{\mathrm{inact}}$ and $\ell_y^{\mathrm{act}}$ be the lower bounds on $y$ obtained after bound propagation in each subproblem. Then the refined lower bound $\ell_y^{\mathrm{new}} := \min\left(\ell_y^{\mathrm{inact}}, \ell_y^{\mathrm{act}}\right)$ satisfies $\ell_y^{\mathrm{new}} \geq \ell_y$, i.e., it is at least as tight as the original.*

*Similarly, the refined upper bound $u_y^{\mathrm{new}} := \max\left(u_y^{\mathrm{inact}}, u_y^{\mathrm{act}}\right)$ satisfies $u_y^{\mathrm{new}} \leq u_y$.*

*As a consequence: If $\ell_y^{\mathrm{new}} \geq 0$, the phase of $z$ is fixed to* active*; and if $u_y^{\mathrm{new}} \leq 0$, the phase of $z$ is fixed to* inactive*.*

*Proof Sketch.* Simulating a split on $z_r$ refines the input domain into two disjoint subdomains. Bound propagation on each subproblem yields tighter constraints on all dependent variables, including $y$. Because the union of the subdomains recovers the full feasible region, the maximum lower bound and minimum upper bound across the subproblems provide valid bounds. $\square$

This phase-fixing capability of lookahead branching not only improves the quality of branching decisions but also reduces the overall search space by pruning redundant subproblems. This makes the branch-and-bound process more efficient and scalable.

## 4 EXPERIMENTAL EVALUATION

To evaluate the effectiveness of lookahead branching, we implemented Algorithm 1 in two state-of-the-art BaB-based verification tools Marabou and $\alpha$-$\beta$-CROWN. These two solvers use different approaches to solve subproblems: Marabou is a CPU-based verifier that employs an SMT-based incomplete decision procedure (Katz et al., 2017; Wu et al., 2022), while $\alpha$-$\beta$-CROWN runs GPU-accelerated bound-propagation (Zhang et al., 2018; Xu et al., 2020). We evaluate lookahead branching against the present state-of-the-art heuristics in each verifier on various benchmarks for each. Our investigation aims to determine if incorporating lookahead branching can improve the performance of branch-and-bound across distinct solver paradigms.

### 4.1 CASE STUDY ON MARABOU

#### 4.1.1 EXPERIMENTAL SETUP

For experimentation on Marabou, we compared lookahead branching to BaBSR and pseudo-impact, two of Marabou's existing heuristics. BaBSR is a widely-used deterministic heuristic (Bunel et al., 2020), while pseudo-impact is a dynamic heuristic specific to Marabou (Wu et al., 2022). We conducted experiments on three different benchmark sets, NAP, NN4Sys, and MNIST. NAP is a benchmark designed to evaluate neural network verifiers on specifications defined by neural activation patterns, which characterize broad, numerically challenging regions of the input space. NN4Sys is a benchmark suite constructed from neural networks used in computer systems, testing real-world verification challenges that have been proven difficult in recent iterations of the VNN Competition. The MNIST dataset includes various feed forward neural networks for handwritten digit recognition. The MNIST network architectures we tested include, in layers by neurons, 20x20, 2x256, 4x256, and 6x256. We implemented lookahead on top of the most recent version of Marabou presented in Wu et al. (2024).

We use the polarity-based pre-selecting strategy as described in Section 3.2 to select 10 candidate neurons. We use a lookahead depth of 2 (parameter **d** in Algorithm 1. We instantiate BASESELECT with two different simpler heuristics, BaBSR and pseudo-impact. In the experiments, we perform lookahead splits on the top five branching levels, and fall back to the simpler heuristics for the rest of it. We used the neuron-fixed-based metric for evaluating a search state and combining the scores across subproblems 3.3. The experiments were run on a server with 2.6-GHz AMD CPUs, with 4 CPU cores allocated per benchmark. Each benchmark was given a 30 minute time limit and a 32 GB memory limit.

### 4.1.2 RESULTS

Table 1 presents a comparison of the performance of Marabou with and without lookahead branching across the various benchmarks. The tables report the number of instances solved within the time limit (1800 seconds), along with average run time on solved instances. Overall, lookahead branching results in a substantial increase in the number of solved instances for both BaBSR and pseudo-impact branching heuristics.

Table 1: Marabou results on all instances

| Benchmark Set | # Benchmarks | babsr | | babsr+lookahead | | pseudo-impact | | pseudo-impact+lookahead | |
|---|---|---|---|---|---|---|---|---|---|
| | | Solved | Avg Time (s) | Solved | Avg Time (s) | Solved | Avg Time (s) | Solved | Avg Time (s) |
| NAP | 235 | 30 | 29.04 | **47** | 3.67 | 19 | 0.54 | **24** | 3.01 |
| NN4SYS | 120 | 80 | 127.12 | **80** | 114.94 | 82 | 149.90 | **94** | 238.13 |
| MNIST 20x20 | 500 | 183 | 100.45 | **192** | 130.62 | 130 | 14.90 | **141** | 9.28 |
| MNIST 2x256 | 500 | **370** | 28.34 | 369 | 30.28 | 405 | 77.92 | **407** | 83.14 |
| MNIST 4x256 | 500 | 281 | 30.09 | **286** | 30.46 | **298** | 72.54 | 295 | 52.19 |
| MNIST 6x256 | 500 | 248 | 86.06 | **254** | 94.01 | 240 | 42.48 | **240** | 40.88 |

Figure 3 provides cactus plots comparing lookahead and the baseline heuristics on the three benchmark sets. Overall, lookahead leads to more solved instances compared to the baseline heuristics, especially when the time limit is high. On the NAP benchmark, the lookahead's overhead makes it slower when the time limit is low, but the improvement in branching quality leads to significantly more solved instances in total. On the NN4Sys benchmarks, lookahead leads to significantly more solved instances in less time with pseudo-impact, but only yields solve time improvements with BaBSR. For the MNIST benchmarks, lookahead leads to an improvement in solved instances when the time limit is higher.

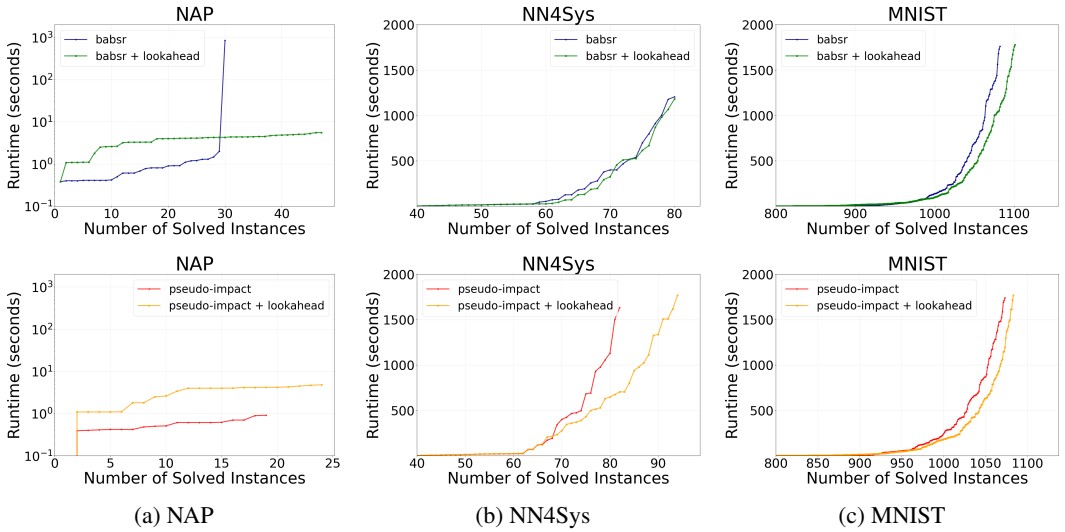

(a) NAP        (b) NN4Sys        (c) MNIST

Figure 3: Cactus plots comparing existing heuristics and lookahead on Marabou

## 4.2 CASE STUDY ON $\alpha$-$\beta$-CROWN

### 4.2.1 EXPERIMENTAL SETUP

For experimentation on $\alpha$-$\beta$-CROWN, we compared lookahead branching to its current state-of-the-art heuristic, FSB(De Palma et al., 2021), on seven different convolutional neural networks ranging with various robustness properties for each. The networks came from the CIFAR, MNIST, and TinyImageNet datasets, three computer vision datasets for image and text recognition. The networks from the CIFAR and TinyImageNet datasets range between 14.4 million and 31.6 million parameters, making them challenging verification queries. The MNIST benchmarks were adversarially trained, introducing complexity that makes formal verification particularly challenging. We

implemented lookahead on top of the most recent version of $\alpha$-$\beta$-CROWN presented in Shi et al. (2025b).

With $\alpha$-$\beta$-CROWN, the solver first attempted to solve the problem with an adversarial attack algorithm(Zhang et al., 2022b), then with incomplete verification using auto-LiRPA(Xu et al., 2021) before beginning complete verification with BaB. When BaB is run, the first five BaB rounds are done using lookahead branching with a lookahead depth of 2, and then FSB branching is used. We use BaBSR scores as the preselect strategy and select 15 candidate neurons. We used the bound-reduction-based metric for the scoring function (Section 3.3) with a discount factor $\lambda$ of 0.5. We did not implement the phase fixing techniques in $\alpha$-$\beta$-CROWN, as $\alpha$-$\beta$-CROWN by default only tightens the bounds of neurons in the final layer. The experiments were run on a server with 2.6-GHz AMD CPUs and A100 GPUs. One A100 GPU and 96 CPU cores were allocated for each experiment, and a 128 GB memory limit was given for each experiment.

### 4.2.2 RESULTS

Table 2 presents a comprehensive overview of the verification performance on the seven neural network models using both FSB branching and lookahead branching within $\alpha$-$\beta$-CROWN. The table compares the total number of solved instances (including both SAT and UNSAT) and the average runtime on solved instances.

Table 2: $\alpha$-$\beta$-CROWN results on all instances

| Dataset | Model | # Benchmarks | Timeout (s) | Solved Instances | | Avg Solve Time (s) | |
| --- | --- | --- | --- | --- | --- | --- | --- |
| | | | | FSB | Lookahead | FSB | Lookahead |
| CIFAR | CIFAR100 | 200 | 360 | 112 | 112 | 19.49 | **16.47** |
| | CIFAR10-ResNet | 72 | 360 | 59 | **60** | 22.83 | **21.87** |
| | CNN-A-Mix | 200 | 200 | 83 | 83 | 7.58 | **6.03** |
| | CNN-B-Adv | 200 | 450 | 93 | 93 | 10.23 | **8.60** |
| MNIST | CNN-A-Adv | 200 | 200 | 141 | 141 | 11.34 | **8.90** |
| TinyImageNet | tinyimagenet | 200 | 360 | 129 | **130** | 23.84 | **20.83** |

To isolate the impact of lookahead branching on the branch-and-bound core of $\alpha$-$\beta$-CROWN, Table 3 presents results filtered to include only instances that required the full BaB procedure. SAT instances are excluded as they were all solved through adversarial attacks, and easier UNSAT solved with incomplete verification methods are also excluded, since lookahead branching has no effect on these cases. We see that lookahead leads to a consistent speedup in solve time, and contributes two unique solutions. Ablation studies on $\alpha$-$\beta$-CROWN are provided in Appendix A. Overall, we found that performance gain can be consistently achieved for various choices of the parameters.

Table 3: $\alpha$-$\beta$-CROWN results on UNSAT instances solved with BaB

| Dataset | Model | Solved Instances | | Avg Solve Time (s) | | Avg Speedup (%) |
| --- | --- | --- | --- | --- | --- | --- |
| | | FSB | Lookahead | FSB | Lookahead | |
| CIFAR | CIFAR100 | 90 | 90 | 23.34 | **19.58** | 16.1% |
| | CIFAR10-ResNet | 19 | **20** | 69.51 | **64.27** | 24.3% |
| | CNN-A-Mix | 32 | 32 | 19.10 | **15.08** | 15.4% |
| | CNN-B-Adv | 47 | 47 | 19.77 | **16.54** | 21.9% |
| MNIST | CNN-A-Adv | 107 | 107 | 14.77 | **11.53** | 16.3% |
| TinyImageNet | tinyimagenet | 116 | **117** | 26.11 | **22.74** | 21.0% |

Figure 4 shows the cactus plots comparing FSB and lookahead for each network. We see that lookahead generally yields faster solve times, especially when the time limit is low. As the time limit increases, FSB tends to catch up.

Overall, we found that lookahead can boost the performance of BaB in two fundamentally different neural network verifiers. This suggests that lookahead branching can be a generally applicable strategy for enhancing the performance of complete neural network verification.

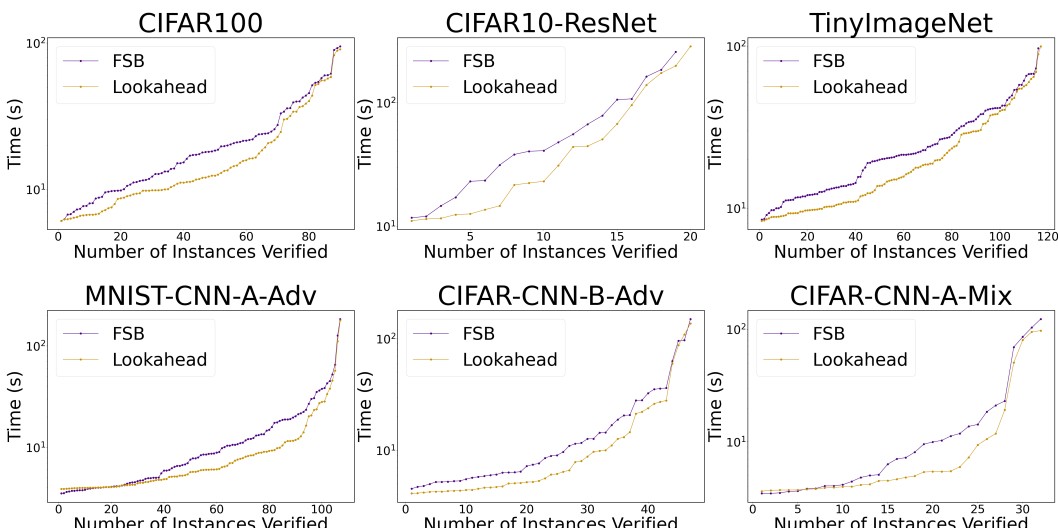

Figure 4: Cactus plots comparing FSB and lookahead on $\alpha$-$\beta$-CROWN

## 5 RELATED WORK

Early efforts for complete verification of neural networks employed SMT and MILP solvers that enumerate activation patterns (Katz et al., 2017). A unified BaB view of verification was presented by Bunel et al. (2020). State-of-the-art complete verification tools including the SMT-based solvers (Katz et al., 2019; Wu et al., 2024) and GPU-accelerated bound-propagation-based tools (Wang et al., 2021; Xu et al., 2020; Zhang et al., 2022a; Zhou et al., 2024; Shi et al., 2025a) can be viewed as instantiation of BaB. Branching heuristics have a significant impact on the runtime of complete verification tools. BABSR introduced a "strong-branching" style score that solves a cheap surrogate relaxation for each candidate and became the default in many verifiers (Bunel et al., 2020). FILTERED SMART BRANCHING (FSB) refines this idea by first filtering candidates with BaBSR and then re-running a tighter bound computation on the shortlist (De Palma et al., 2021). There has also been work on using Graph Neural Networks to train a policy for selecting splitting neurons (Lu & Kumar, 2019). Our work is inspired by *lookahead* search in SAT and SMT solvers (Heule & van Maaren, 2009) as well as MILP solvers (Glankwamdee & Linderoth, 2006), and we extend similar ideas to neural network verification.

## 6 CONCLUSION

In this paper, we introduced a lookahead based branching strategy for neural network verification. The key idea is to simulate candidate splits prior to committing to a split in order to make more informed branching decisions. We discussed design choices of lookahead and instantiated lookahead branching for two distinct complete verifiers, Marabou and $\alpha$-$\beta$-CROWN. Our results show that incorporating lookahead branching results in substantial performance gains in both solvers across a wide range of benchmarks. This suggests that lookahead is a generally applicable strategy for neural network verification.

**Limitations** As we presented lookahead as a template algorithm, its design space is massive. While we considered two instantiations of lookahead in Marabou and one in $\alpha$-$\beta$-CROWN, it would be interesting to evaluate more variants of lookahead. In particular, we plan to investigate the effect of performing simulations to deeper levels or periodically invoking lookahead later in the search. In addition, while we explored leveraging lookahead to fix unstable neurons, it might be interesting to leverage other information, such as dependencies between neurons, which might result in additional pruning of the search space.

## 7 ETHICS STATEMENT

LLMs were used in the process of paper writing to assist with phrasing and grammar. Specifically, LLMs were used to improve clarity, refine phrasing, and ensure consistency in language. All methodology, experimental results, and analyses were written by the authors, and no text was generated without author oversight. We acknowledge this usage in accordance with ICLR's ethical guidelines and take full responsibility for the integrity and accuracy of the submission.

## 8 REPRODUCIBILITY STATEMENT

The full source code for both Marabou and $\alpha$-$\beta$-CROWN is included in the supplementary materials, along with instructions for replication. Hardware specifications, experiment configurations, and hyperparameters are documented in Section 4. All benchmark datasets are publicly available and used without modification.

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

# A    ABLATION STUDIES ON $\alpha$-$\beta$-CROWN

Table 4: $\alpha$-$\beta$-CROWN results with varying number of lookahead branches

| Dataset | Model | 1 Lookahead Branch | | 5 Lookahead Branches | | 10 Lookahead Branches | |
|---|---|---|---|---|---|---|---|
| | | Solved | Avg Time (s) | Solved | Avg Time (s) | Solved | Avg Time (s) |
| CIFAR | CIFAR100 | 90 | 64.21 | 90 | 64.27 | 90 | **57.25** |
| | CIFAR10-ResNet | 20 | 20.40 | 20 | **19.58** | 20 | 21.49 |
| | CNN-A-Mix | 32 | 16.15 | 32 | 15.08 | 32 | **14.99** |
| | CNN-B-Adv | 47 | 17.24 | 47 | **16.54** | 47 | 16.86 |
| MNIST | CNN-A-Adv | 107 | 11.89 | 107 | **11.53** | 107 | 13.65 |
| TinyImageNet | tinyimagenet | 117 | 24.66 | 117 | **22.74** | 117 | 24.93 |

Table 4 presents the performance of $\alpha$-$\beta$-CROWN while varying the number of lookahead branches performed before switching to a simpler heuristic. Across all lookahead depths tested, lookahead either solves more instances or has faster solve times than FSB. Changing the lookahead depth does not change the number of solved instances, but has a non-trivial impact on the average solve time.

