# OpenReview forum: "Lookahead Branching for Neural Network Verification"
_ICLR.cc/2026/Conference — ICLR 2026 Conference Withdrawn Submission_

### Official Review · Reviewer_Rb31 · 2025-10-16

**Soundness:** 3
**Presentation:** 3
**Contribution:** 2
**Rating:** 4
**Confidence:** 4

**Summary:**

This paper introduces a heuristic branching strategy named "Lookahead Branching", aimed at improving the efficiency of neural network verification. The core idea is to enhance the Branch-and-Bound (BaB) process by making more informed branching decisions. This is achieved by first ranking candidate neurons based on certain metrics, selecting a subset for lookahead, and then using a scoring function to evaluate the potential gain of splitting on a specific neuron. Furthermore, the method can discover and leverage information during the simulation to tighten variable bounds, thereby pruning the search space. The authors present this strategy as a general framework and successfully instantiate it in two different types of verifiers: Marabou (an SMT-based solver) and α,β-CROWN (a bound-propagation-based verifier). Experimental results demonstrate that this approach leads to performance improvements across various benchmarks.

**Strengths:**

* The branching strategy is a critical factor influencing the performance of BaB-based verifiers. The paper's focus on investing more upfront computational effort to achieve better global search efficiency is a well-motivated and valuable research direction.
* The paper successfully implements the lookahead strategy in two verifiers with distinct architectures—Marabou, an SMT-based CPU solver, and α,β-CROWN, a GPU-accelerated solver based on bound propagation—and achieves positive results in both.
* The experiments show that, compared to existing state-of-the-art heuristics, the lookahead strategy achieves notable improvements on several standard benchmarks, both in terms of the number of instances solved (for Marabou) and the solve time (for α,β-CROWN).

**Weaknesses:**

Despite the promising research direction and initial results, I have the following concerns:

* Insufficient analysis of parameter selection and sensitivity: The lookahead strategy is presented as a "template algorithm" with multiple hyperparameters, such as the lookahead depth d, the size of the pre-selected candidate set, the number of search levels for its application, and the discount factor λ in the scoring function. The paper does not provide a detailed explanation of how these key parameters were chosen, nor does it include a thorough sensitivity analysis. For instance, what is the rationale for setting the lookahead depth to 2? How do deeper or shallower depths affect performance and overhead? While the appendix offers a small ablation study on the number of branching levels, this is hardly sufficient to cover the vast design space. This makes it difficult for readers to judge the extent to which the method's success relies on meticulous parameter tuning and undermines its "plug-and-play" utility as a general method.
* Diminishing returns on larger networks: The benefits of the proposed method appear to diminish as the network size increases. This is evidenced by the marginal difference in the number of solved instances (sometimes just one more instance is solved) and the lack of significant improvement in average solve time (e.g., the difference is around 1 second on some benchmarks). This raises concerns about the scalability and effectiveness of the method on larger, real-world-scale networks.
* Limited novelty: The paper acknowledges that the concept of lookahead search is well-established and has been extensively applied in fields such as SAT/SMT solvers. Therefore, the main contribution of this work lies in the successful adaptation of this mature idea to the domain of neural network verification. While this adaptation work is valuable, the fundamental novelty of the approach is relatively limited.
* There appear to be some inconsistencies in the experimental data (as detailed in the Questions section).

**Questions:**

* How should the hyperparameters for this method be selected? Is fine-grained tuning necessary for different network architectures or verification problems?
* What proportion of the total runtime is consumed by the lookahead decision process itself? Can an analysis be provided on how this proportion changes with problem difficulty and network scale?
* The proposed method shows significant improvement on UNSAT instances (Table 3), but the overall average solve time on some datasets does not differ much (Table 2). Does this suggest that the method introduces some performance overhead on SAT instances?
* How is the "Avg Speedup" in Table 3 calculated? Some figures seem inconsistent. For example, for CIFAR10-ResNet in Table 3, the original average solve time is 69.51s and the improved time is 64.27s, which does not seem to correspond to the claimed 24.3% speedup.

Additionally, there are some minor writing issues:

* Page 1, line 31: 'has significant impact' is missing an article (should be 'has a significant impact').
* Page 3, line 145: Typo, 'evalutes' should be 'evaluates'.
* Page 7, line 373: The phrase 'ranging with' appears to be used incorrectly.
* The tool α-β-CROWN is more conventionally written as α,β-CROWN.
* There is an inconsistency in terminology, with both 'neuron-fixed-based' and 'neuron-fixing-based' being used.

---

### Official Review · Reviewer_Huo6 · 2025-10-19

**Soundness:** 3
**Presentation:** 3
**Contribution:** 2
**Rating:** 2
**Confidence:** 4

**Summary:**

Branch-and-Bound (BaB)-based verifiers which make use of cheap linear relaxations together with sophisticated branching strategies have emerged to be the best-performing neural network verifiers in recent year. The strategy that is used for making branching decisions is an important part of these algorithms since inefficient branching may lead to significantly increased verification times. Presently, Filtered Smart Branching (FSB) is the most popular branching heuristic due to its effectiveness which is achieved by simulating the effects of different branching decisions. This work proposed lookahead branching which, unlike FSB, considers the impact of branching decisions at greater depths before making a branching decision. A number of neurons are first selected according to a preselection strategy. A more detailed analysis is then conducted for those neurons and scoring functions are used to summarise the results of the simulations. Based on the obtained results, a branching decision is then made. The authors implement their method in two popular neural network verification tools (Marabou and $\alpha, \beta$-CROWN) for testing. The empirical evaluation demonstrates that they lead to more verification queries being resolved and shorter verification times.

**Strengths:**

- The investigation of branching heuristics is an important research topic since they are a key component of neural network verifiers and have a significant impact on their effectiveness
- The experiments seem promising, showing that the improved heuristics accelerate neural network verification
- The two neural network verifiers that are used in the evaluation section are quite different in how they work
- The work is well-motivated

**Weaknesses:**

- Given that the main topic tackled in this work is that of branching heuristics in neural network verifiers, I was surprised that the background section of the paper does not contain any information about existing branching heuristics. The baseline heuristics (FSB, BaBSR) should be properly introduced and explained in the paper. At the moment e.g. FSB is also mentioned multiple times before it is even introduced in line 463, so the paper needs some reorganisation of its contents such that concepts are explained before being referred to.
- The paper lacks a proper comparison between the proposed heuristics and the existing ones. As far as I'm aware, FSB also simulates the impact of branching decisions before branching, but only runs the simulation at a branching depth of $1$. The main difference between this and the heuristic presented in this work seems to be that the authors run the simulation at greater branching depths which is a straightforward extension of FSB and, as noted by the authors, has been proposed in a variety of other fields such as MILP solving.
- The selection of hyperparameters is unclear to me. As an example, the authors use $10$ candidate neurons for Marabou and $15$ for $\alpha, \beta$-CROWN but never explain this choice. An ablation study should be added to investigate the impact of different numbers of candidates on the effectiveness of the method. They also use lookahead for the *top five branching levels* in Marabou while conducting the *first five BaB rounds* with lookahead in $\alpha, \beta$-CROWN - why are different strategies used here?
- The Marabou experiments do not compare the proposed method to the SoA FSB heuristic, but only to the older BaBSR heuristic. The comparison would be more convincing if the authors could compare their proposed method to FSB on Marabou.
- In Theorem 1 the authors take the minimum of the lower bounds and the maximum of the upper bounds to be the "refined" bounds after branching. This contradicts the proof sketch which mentions "the maximum lower bound and minimum upper bound across the subproblems" should be used as the new valid bounds. Could the authors clarify which of the formulations is correct?

**Questions:**

- Why do the authors use a lookahead depth of $2$?
- Why are different hyperparameters used for the two verifiers and how were those chosen? The ablation study in the appendix also seems to indicate that different numbers of lookahead branches seem optimal for different benchmarks, how would the authors suggest the various hyperparameters of the method be selected for new benchmarks?
- What is the technical contribution of the work over existing heuristics such as FSB or the one presented in the paper by Botoeva et al. that the authors cite?

---

### Official Review · Reviewer_c9QU · 2025-10-21

**Soundness:** 1
**Presentation:** 2
**Contribution:** 3
**Rating:** 2
**Confidence:** 5

**Summary:**

This paper presents a new method for choosing splits in branch and bound neural network verification, based on examining the branching tree until a fixed depth. This lookahead strategy for split selection is used during the first steps of the branch and bound process. When integrated in the Marabou and alpha-beta-CROWN verifiers, it allows them to handle more benchmark problems and to decrease the total solving time.

**Strengths:**

This paper proposes a new branching heuristic that shows good results in the experiments. Because the heuristic is somewhat costly, it is used only during the first few iterations of the branch-and-bound process. In this way, it gives a kind of “warm start” to the verifier, which is a fresh and interesting idea for neural network verification. The presentation is, for the most part, satisfactory.

**Weaknesses:**

### Mayor Comments
1. The proof of Theorem 1 is not correct, I am afraid. The state-of-the-art bounding methods CROWN and alpha-CROWN do not enjoy the monotonicity property assumed in the argument. With the adaptive ReLU rule, CROWN bounds can even become looser when the pre-activation bounds are tightened. And since alpha-CROWN optimises the bounds by gradient descent, these bounds may change in an essentially arbitrary way at each invocation.
2. The reported runtimes lack standard deviations or inter-quartile ranges. As the results are all in a similar range, the measurements should be repeated several times. In the ideal case, the paper would report the median runtimes together with their inter-quartile ranges.

### Minor Comments
3. Some citations are missing, for instance for VNNComp.
4. Presentation:
    4a. The verification problem is not clearly defined. It is not explained what the formula $\phi$ represents in a formal sense.
    4b. The description of branch and bound omits a key aspect: some branches created by ReLU-splitting can be infeasible.
    4c. The phrase « most undecided » in line 219 is not clear. It should indicate the precise criterion.
    4d. The expression « time limit is higher » in line 247 is unclear in what it refers to.
    4e. The description of MILP in Section 5 is not correct. A MILP formulation does not enumerate activation patterns.
    4f. It is unclear why the paper applies a discount factor to the bound-based metric but omits it for the neuron-fixing metric.

## Typographical Errors
- Line 11 should read: « the effect of *a* lookahead branching strategy ».
- There should not be a line break just below the headline of Section 2.3.
- There should be more space below Figure 2.
- Please clarify the notation in Algorithm 1, or replace it with a more explicit form. For instance, the meaning of the operation « $S::S'$ is not explained.
- The word « Section » is missing in line 322.
- There is a missing space between « FSB » and the citation in line 373.
- Line 381 should use the present tense for consistency with the rest of the paper.

**Questions:**

1. The Appendix provides results for a lookahead of one. How do these differ from filtered smart branching?

Overall, the paper explores an interesting new direction for neural network verification. However, the theoretical results are not correct, and the experimental evaluation needs more reliable runtime measurements to support the paper’s claims. Beyond the incorrect proof, any theoretical result should be presented with a complete proof, not only a sketch. Going forward, it may be clearer to present the new split selection heuristic as a warm-start strategy for the verifier, highlighting that its main benefit lies in the early iterations of branch and bound.

---

### Official Review · Reviewer_5k8Q · 2025-10-30

**Soundness:** 2
**Presentation:** 3
**Contribution:** 1
**Rating:** 2
**Confidence:** 5

**Summary:**

This paper proposes lookahead branching for neural network verification. They presenting the approach as a general template algorithm that can be integrated into the popular branch-and-bound (BaB) frameworks. The idea involves simulating potential branching decisions by performing multiple splits to measure downstream effects before committing to a split.  The authors instantiate their method in two verification tools: Marabou (using neuron-fixing-based scoring) and ABCrown (using bound-reduction-based scoring).  Evaluation demonstrates performance improvements on various benchmarks.

**Strengths:**

- Txperiments show measurable improvements in both verification systems, e.g., Marabou showing increased solved instances and ABCrown achieving 15-25% average speedups on BaB-required problems.

- The authors demonstrate implementation across two different verification paradigms (SMT-based Marabou and GPU-accelerated ABCrown), showing the approach can be adapted to different solver architectures.

- The paper provides a structured template framework with clearly defined components (preselect, simulate, evaluate, aggregate) that could potentially guide implementation in other verifiers.

**Weaknesses:**

- The proposed algorithm is a straightforward application of combining greedy best-first search with limited look-ahead depth, a standard practice in MILP solvers, constraint programming, and game (minimax with look-ahead).

- The proposed algorithm appears similar to existing methods, FSB heuristic, already implemented in ABCrown (see `alpha-beta-CROWN/complete_verifier/heuristics/fsb.py`):
     - Candidate preselection: both use top-k candidate selection
     - Scoring functions: both leverage BaBSR scores for candidate ranking (using different scores sound too incremental)
     - Depth-limited search: both explore candidates up to a specified depth
     - Split simulation: the `SIMULATE_SPLIT` is similar to FSB's `build_history_and_set_bounds` functionality

- The approach exhibits several inconsistencies:
     - Different scoring functions used for verifiers (neuron-fixing for Marabou vs. bound-reduction for ABCrown)
     - Different benchmark sets used for verifiers
     - Paper claim a significant outcome of the lookahead branching procedure is `phase fixing`, but later on (L385-386), `we did not implement the phase fixing techniques in ABCrown`.

- The experimental results show only marginal gains:
     - Except NAP, Marabou shows minimal increases in solved instances
     - ABCrown shows average speedups of 15-25% but the same number of solved instances in most cases (sometimes worse)
     - Parameters are chosen without justification, lacks of evaluating performances with parameter tuning

- Minor: some procedures from main algorithm have not been discussed/defined (applySplit, propagateBounds, etc.)

**Questions:**

- What are the conceptual differences between the proposed method and the existing FSB implementation in ABCrown?

- Why do you use different scoring functions and benchmarks for different verifiers? How do these affect the generalizability claims of the template framework?

- The paper claims phase fixing as a significant outcome but admits not implementing it in ABCrown. How can the contribution be fairly evaluated without this key component?

---

### Note · Authors · 2025-11-12

I have read and agree with the venue's withdrawal policy on behalf of myself and my co-authors.